# Clinical Significance of Adenosine-Induced Atrial Fibrillation after Complete Pulmonary Vein Isolation

**DOI:** 10.3390/jcm11195679

**Published:** 2022-09-26

**Authors:** Yun Young Choi, Jaemin Shim, Yun Gi Kim, Kyongjin Min, Seung-Young Roh, Jin Seok Kim, Jong-Il Choi, Young-Hoon Kim

**Affiliations:** 1Department of Internal Medicine, Graduate School, Kyung Hee University, Seoul 02447, Korea; 2Division of Cardiology, Korea University College of Medicine and Korea University Medical Center, Seoul 02841, Korea; 3Korea University Anam Hospital, Seoul 02841, Korea

**Keywords:** atrial fibrillation, adenosine, recurrence, radiofrequency catheter ablation

## Abstract

Background: Adenosine can cause dormant electrical conduction between the pulmonary vein and left atrium after pulmonary vein isolation (PVI). Adenosine can also induce atrial fibrillation (AF) during catheter ablation. However, the clinical outcomes and effects of additional ablation for the trigger sites of adenosine-induced AF (AIAF) are unknown. This study therefore aimed to evaluate the clinical significance of AIAF. Methods: Between January 2010 and September 2019, we analyzed 616 consecutive patients with paroxysmal AF (PAF) who underwent radiofrequency catheter ablation (RFCA), including wide-area circumferential pulmonary vein isolation (PVI) and post-PVI adenosine testing. Results: Among 616 patients, 134 (21.7%) and 34 (5.5%) showed dormant conduction and AIAF, respectively. Eight patients (1.3%) had both dormant conduction and AIAF. The AF recurrence rate was not significantly different between patients with and without AIAF (16.7% vs. 18.6%, log-rank *p* = 0.827) during a mean follow-up period of 17.9 ± 18 months. Additional RFCA for the trigger site was attempted in 10 patients with AIAF; however, the recurrence rate of atrial arrhythmias was also not different between the groups with and without additional ablation (20% vs. 16.7%, log-rank *p* = 0.704). Conclusions: AIAF after PVI was not clinically associated with recurrence during long-term follow-up. Ablation of the trigger site in AIAF did not improve the clinical outcomes.

## 1. Introduction

According to the 2020 European Society of Cardiology (ESC) guidelines, radiofrequency catheter ablation (RFCA) is beneficial for patients with symptoms of atrial fibrillation (AF) and those refractory to medical treatment [1]. It is known that up to 14% of patients who undergo RFCA for AF experience complications [2]; however, this rate has been reported to be gradually decreasing owing to the recent development of ablation technologies and appropriately trained operators. RFCA for AF has been proven to improve the quality of life compared to antiarrhythmic drug administration [3], but the 1-year success rate of patients undergoing pulmonary vein isolation (PVI) for paroxysmal AF (PAF) has been reported to be 50–80% [4]. Therefore, it is important to reduce the recurrence of atrial arrhythmia by identifying risk factors for recurrence. Several studies have shown that a history of hypertension, impaired renal function, and diabetes mellitus, and among echocardiographic risks, decreased ejection fraction and enlarged LA diameter are associated with an increased risk of recurrence [5,6].

Recurrence may also occur during RFCA due to the electrical recovery of the initially isolated PV and atrial arrhythmias originating from causes other than the pulmonary veins. Adenosine injection has been used to confirm non-inducibility or unmask dormant PV-LA conduction. Adenosine may uncover non-PV lesions after antral PVI in >10% of patients [7]. A meta-analysis of six studies showed that routine adenosine testing was associated with a reduction in AF recurrence after PVI [8]. The elimination of non-PV triggers and dormant conduction has been demonstrated to be effective in reducing recurrence as the endpoint of the procedure [8,9]. However, it is unclear whether the ablation of adenosine-induced AF (AIAF) can reduce the recurrence of atrial arrhythmias. AIAF of non-PV origin is approximately 5% of patients with PAF [10]. However, the correlation between AIAF and dormant conduction is not well-known, and the clinical relevance of AIAF has not been well studied. Therefore, this study aimed to investigate the clinical impact of additional ablation of AIAF on the reduction in recurrence in patients with PAF.

## 2. Methods

### 2.1. Study Population and Design

We retrospectively analyzed all consecutive patients who underwent RFCA for PAF at the Korea University Medical Center between July 2010 and September 2019. The exclusion criteria were as follows: (1) Patients who underwent RFCA for persistent AF, (2) patients who did not receive adenosine administration during the procedure, (3) contraindication to anticoagulation, (4) history of any precious AF ablation, (5) presence of intracardiac thrombus, and (6) pregnancy. The Institutional Review Board of the Korea University Anam Medical Center approved the study protocol. All eligible participants provided written informed consent prior to enrolment. PAF was defined as the documentation of at least one AF episode lasting ≥ 30 s on 12-lead electrocardiography or 24 h Holter monitoring and terminating either spontaneously or by intervention within 7 days of onset [1]. Antiarrhythmic medications were discontinued for 1 week and oral anticoagulation was maintained for at least 4 weeks before the procedure.

### 2.2. Electrophysiologic Studies and Catheter Ablation

All patients underwent transthoracic echocardiography (GE Vivid E9; Vingmed Ultrasound, Horten, Norway) prior to RFCA and, if necessary, transesophageal echocardiography or intracardiac echocardiography to rule out left atrial (LA) thrombus.

An electrophysiological study was conducted on patients who were sedated by an intravenous injection of midazolam and propofol, and arterial blood gas analysis was performed hourly. Intracardiac electrocardiograms of the coronary sinus (CS) and low right atrium (RA) were mapped with a duodecapolar catheter (St. Jude Medical, Inc., Minnetonka, MN, USA), and the high RA was mapped with a decapolar catheter (Bard Electrophysiology, Inc., Lowell, MA, USA). Two transseptal punctures were performed under fluoroscopic guidance using a Brockenbrough needle and two long sheaths (Fast-CathTM and SwartzTM SL1; St Jude Medical, AF Division, Minnetonka, MN, USA). Unfractionated heparin was injected to maintain an activated clotting time within 300–350 s. After two transseptal punctures, an ablation and mapping catheter was inserted, and cardioversion was performed to restore sinus rhythm. Three-dimensional electro anatomical mapping was performed using NavX (St. Jude Medical, St. Paul, MN, USA) and CARTO (Johnson & Johnson Inc. Diamond Bar, CA, USA) and merged with cardiac magnetic resonance imaging and spiralComputed Tomography.

All patients underwent wide antral circumferential PVI. Complete PVI was defined as the elimination of antral potentials or dissociation of the PV recorded on a circumferential PV catheter. Additional substrate modifications, defined as additional ablation other than PV isolation, were performed to achieve non-inducibility at the operator’s discretion. The ablation strategy was performed as previously described [9]. We evaluated the non-PV triggers of AF during catheter ablation for PAF and targeted them after PVI. Non-PV trigger of AF was induced by intravenous infusion of high-dose isoproterenol (10 μg per minute) after completion of PVI. If AF was induced, multiple electrical cardioversions were performed to restore sinus rhythm, and the initiation of AF trigger was confirmed. Additional ablation was performed, when non-PV trigger was identified. LA linear ablation was performed in cases where AF was converted to AT or when AT was induced. The endpoint of the procedure for patients with PAF was the elimination of all triggers, including PVs. Cavotricuspid isthmus linear ablation was performed in a patient with clinical atrial flutter; in this case, the endpoint was a bidirectional block by differential pacing.

### 2.3. Adenosine Testing

First, adenosine was administered at 9 or 12 mg and titrated until at least one P wave was blocked or until there was a sinus pause of 3 s or more. Adenosine administration resulted in unmasking of dormant conduction or AIAF between the PV and the LA. Dormant conduction was defined as transient representation of the PV potential recorded on a circumferential PV catheter. AIAF was defined as AF induced after the P wave was blocked or paused after adenosine administration. Adenosine testing was performed until all the PVs demonstrated complete PVI. PVs with dormant conduction were considered reconnected, and further ablation was performed in all cases with dormant conduction. Repeated adenosine administrations were performed to determine the trigger site of earliest activation in the CS and RA electrograms and to confirm the trigger sites by repositioning the circular mapping catheter. The triggered site of earliest activation was eliminated until AIAF did not occur. For cases where the vein of Marshall (VOM) was identified to be the trigger, a perimitral isthmus ablation was performed.

### 2.4. Post Ablation Follow-Up and Definition of Recurrence

The patients were discharged within 3 days of RFCA and visited the outpatient clinic 2 weeks after discharge. If the patient was found to have palpitations, we performed an electrocardiogram (ECG), 24-h Holter monitoring, or event recorder to evaluate the cause. All patients had routine follow-up outpatient clinic appointments at 1, 3, 6, and every 6 months thereafter. ECG was performed at each outpatient visit, and 24-h Holter monitoring was performed at 3, 6, 9, and 12 months after RFCA. Atrial arrhythmia lasting more than 30 s during the first 3 months of the blanking period was defined as early recurrence, and atrial arrhythmia lasting more than 3 months after the procedure was defined as late recurrence. Antiarrhythmic medications were taken for 3 months after RFCA and discontinued at the physician’s discretion.

### 2.5. Statistical Analysis

Continuous variables are expressed as mean ± standard deviation, and unpaired *t*-tests were used to compare continuous variables between the two patient groups. Categorical variables were compared using the chi-squared test or Fisher’s exact test. Kaplan–Meier survival curve analysis was used to delineate the degrees of freedom of recurrence, and log-rank tests were applied to compare differences between groups. Univariate and multivariable Cox regression analyses were performed. Hazard ratios (HRs) and 95% confidence intervals (CIs) were calculated for each independent variable. Two-tailed *p*-values were reported, and the results were considered statistically significant at *p* < 0.05. Statistical analyses were performed using SPSS statistical software (version 24.0; SPSS Inc., Chicago, IL, USA).

## 3. Results

### 3.1. Baseline Characteristics

In total, 616 patients who received adenosine after PVI were enrolled in this study. The baseline characteristics of two groups of patients with and without AIAF are presented in Table 1. The mean age of both groups was 53.9 years and 56.2 years, respectively, and males were dominant in both groups. Left atrial diameter and body mass index were significantly lower in the AIAF group than in the non-AIAF group. However, both values were within normal ranges.

### 3.2. Dormant Conduction and Adenosine-Induced Atrial Fibrillation

AIAF was identified in 34 patients (5.5%). Negative dormant conduction was observed in 76.5% of patients with AIAF and 78.4% of patients without AIAF. Among the patients with AIAF, 8 (23.5%) showed dormant conduction. Dormant conduction was observed in 23.5% of patients with AIAF and 21.6% of patients without AIAF, which was not significantly different. Additional ablation for triggers of AIAF was performed in 10 of 34 patients (Figure 1). The trigger sites for AIAF are shown in Figure 2. In half of the cases, CS and RA catheters or circular mapping catheters indicated that the trigger sites were in the RA, but the detected anatomical location was not specified due to non-persistence or lack of reproducibility. The trigger sites where AIAF occurred were 35% in the crista terminalis (CT), 6% in the RA appendage (RAA), and 3% in the CS ostium, right septum, and VOM. Among them, the ablation sites were CT in seven cases, RAA in two cases, and VOM in one case. Among the CT ablation sites, atrial arrhythmia recurred in two patients (Appendix A). Figure 3 shows additional ablation of the trigger site of AF induced by adenosine. If AIAF occurred in the RAA, the circular mapping catheter was repositioned at the earliest activation site and eliminated until no AIAF occurred. The procedure, ablation, and fluoroscopic time were not significantly different between the AIAF group and patients without the AIAF group (Appendix A).

### 3.3. Recurrence of Atrial Arrhythmia after RFCA

The recurrence rate was not significantly different between patients with and without AIAF (16.7% vs. 18.6%, log-rank *p* = 0.827) (Figure 4A) during a mean follow-up period of 17.9 ± 18 months. In the AIAF group, there was no significant difference in the recurrence rate of atrial arrhythmias between the groups with and without ablation for the trigger site of AIAF (20% vs. 16.7%, log-rank *p* = 0.704) (Figure 4B). The recurrence rates were not significantly different between the non-AIAF group and the group without ablation (18.6% vs. 16.7%, log-rank *p* = 0.740) (Figure 4C). The Cox regression analysis of the risk factors for AF recurrence is shown in Table 2. In univariate analysis, LA diameter ≥ 40 mm (hazard ratio [HR] 1.64, 95% confidence interval [CI] 1.12–2.41, *p* = 0.011) was associated with an increased risk of recurrence of atrial arrhythmia. However, sex, age, comorbidities, and AIAF were not significantly associated with recurrence after RFCA for PAF. Multivariable analysis showed that LA diameter ≥ 40 mm (HR 2.10, 95% CI 1.36–3.27, *p* = 0.001) was also independently associated with recurrence after adjusting for age, sex, body mass index, congestive heart failure, hypertension, diabetes, history of stroke or TIA, vascular disease, AIAF, LA diameter, and LVEF. A total of 54 (8.8%) patients underwent redo-RFCA during the follow-up period. Among them, 2 (5.9%) AIAF patients underwent redo-RFCA.

## 4. Discussion

The main results of this study are as follows. (i) Most of the trigger sites where AIAF occurred were detected in crista terminalis, and all sites except VOM were in RA. (ii) There was no significant difference in the recurrence rate between the AIAF and non-AIAF groups. (iii) In the AIAF group, ablation of the trigger site did not affect AF recurrence. An LA diameter > 40 mm was associated with recurrent AF, whereas AIAF was not significantly associated with recurrence of atrial arrhythmia.

A recent study has shown no significant difference in recurrence between patients with and without AIAF [11]. In addition to previous study, the recurrence rate did not significantly differ between those with and without additional ablation in our study, indicating that additional ablation was not meaningful even in the presence of AIAF. Additionally, the fact that most of the trigger sites were RA can be seen as a difference from previous work.

### 4.1. Adenosine-Induced Atrial Fibrillation and Right Atrium

Adenosine is known to cause membrane hyperpolarization in cardiomyocytes through the activation of adenosine A1 receptors (A1R) and activation of the outward potassium current, I_KAdo,_ formed by G protein-coupled inwardly rectifying potassium channels (GIRK), which shortens the atrial action potential duration (APD) and refractory period [12,13]. The ionic interaction resulting from A1R and GIRK4 protein expressions may contribute to a substrate favorable for atrial arrhythmic activity. In our study, most of the trigger sites where AIAF occurred were the RA. Li et al., showed that A1R and GIRK4 protein expression was three-fold higher in RA than in LA, indicating that RA was significantly more sensitive to repolarization in response to adenosine [14]. Thus, sustained AIAF is mostly maintained by localized re-entrant drivers in the RA region. In addition, A1R and GIRK4 protein expression was higher in the lateral RA than in other atrial regions. In our study, 35% of AIAF cases were detected on CT.

At baseline characteristics, the LA diameter of patients with AIAF was significantly smaller than that of patients without AIAF (37 ± 4.5 vs.39.0 ± 5.3, *p* = 0.014). However, in the Cox regression analysis, there was an association with the recurrence rate when the LA diameter was more than 40 mm. It means that in case of adenosine induced AF electrical properties of atrial myocardium play more important role in AF development than structural changes while in case of recurrent AF vice versa.

### 4.2. Sympathetic Excitatory Effect

Sunny et al., suggested that acute autonomic remodeling induced by the administration of parasympathomimetic agents such as adenosine results in either easily induced or spontaneous sustained AF [15]. After adenosine administration, a sinus pause may occur, followed by AF. This suggests that there is a primary increase in adrenergic effect and marked regulation of vagal nerve dominance before AF and that there is an autonomic remodeling with AF [16]. The vagal response was better induced by adenosine and readily induced AF, with a shorter atrial refractory response due to shorter atrial APD and GIRK current activation. Thus, adenosine administration can promote significant sympathetic effects by activating afferent nerves, primarily via arterial baroreceptors or chemoreceptors [17]. Conversion to vagal nerve dominance due to adenosine administration may increase sympathetic tone, which can promote the induction of AF at arrhythmogenic sites. This study also showed that a long pause was observed after the administration of adenosine, after which spontaneous AF was initiated (Appendix A). We analyzed the correlation between adenosine-induced bradycardia and AIAF. There are no significant differences in the degree of bradycardia (maximal duration of pause) between the two groups.

### 4.3. Comparison to Isoproterenol for Confirming Non-PV Trigger

Another study showed that when isoproterenol and adenosine were administered to the same patients, isoproterenol mainly induced a PV trigger, whereas adenosine-induced a non-PV trigger, especially in RA. However, the trigger site induced by isoproterenol was correlated with recurrence, whereas the non-PV trigger induced by adenosine did not show a clear correlation with recurrence [18]. In addition, isoproterenol is more useful than adenosine when non-PV triggers are induced with isoproterenol and adenosine [19].

### 4.4. Clinical Implications

In our study, no significant difference was observed in the recurrence of atrial arrhythmias with or without AIAF. The adenosine-induced non-PV trigger may be caused by adenosine A1R and GIRK current activation in RA, and AF may occur due to vagal response by enhancing the adrenergic effect. Furthermore, there was no significant difference in the recurrence rate between the group in which AF was not induced by adenosine and the group that was induced but not eliminated. Therefore, eliminating AIAF as an endpoint may be unnecessary.

### 4.5. Limitation

This study has several limitations. First, this is a retrospective single center study and the number of patients enrolled in the study was relatively small. Second, the origin of the trigger site could not be identified in approximately half of the AIAF patients. Third, asymptomatic AF may be missed due to its paroxysmal nature. However, similar to previous studies including patients with PAF, all patients had regular Holter monitoring as well as ECGs at each outpatient visit.

## 5. Conclusions

AIAF most frequently occurred in RA; among these, most occurred in the CT. AIAF was not clinically relevant during long-term follow-up when compared with patients with non-induced AF due to adenosine. Additional ablation of the trigger sites of AIAF did not improve clinical outcomes.

## Figures and Tables

**Figure 1 jcm-11-05679-f001:**
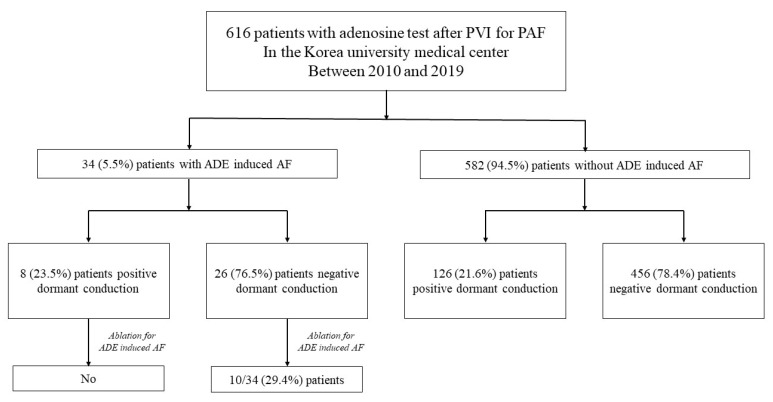
Flowchart showing the study design. PVI = Pulmonary vein isolation; PAF = paroxysmal atrial fibrillation; ADE = adenosine.

**Figure 2 jcm-11-05679-f002:**
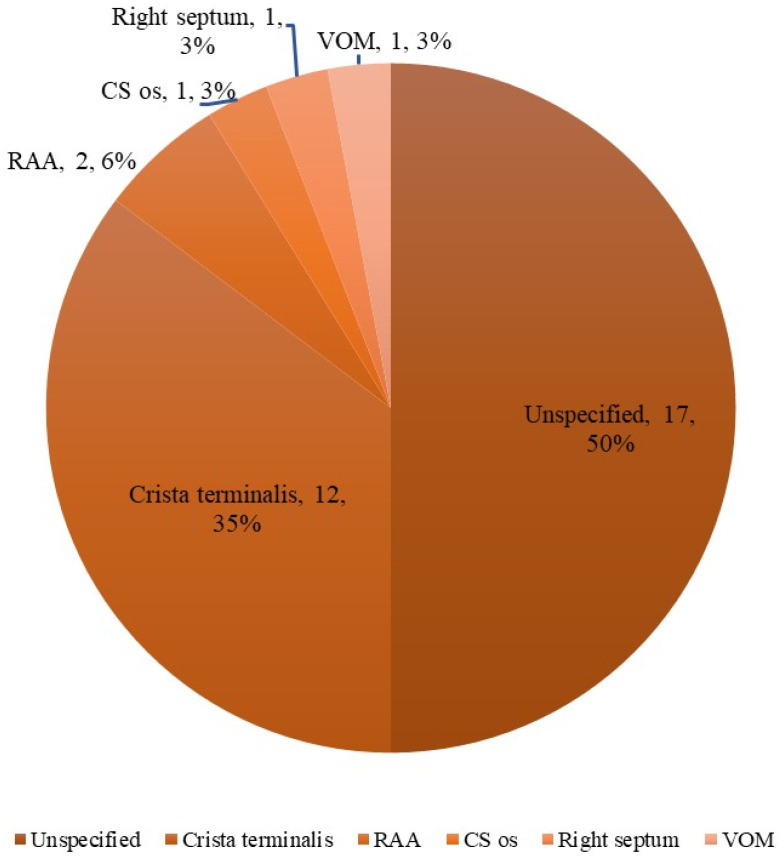
Trigger sites of adenosine-induced atrial fibrillation and recurrence rate after ablation of trigger site. RAA = right atrial appendage; CS os = crista terminalis ostium; VOM = vein of Marshall unspecified. The trigger sites were unspecified, but CS and RA catheters or circular mapping catheters indicated that RA was the trigger site.

**Figure 3 jcm-11-05679-f003:**
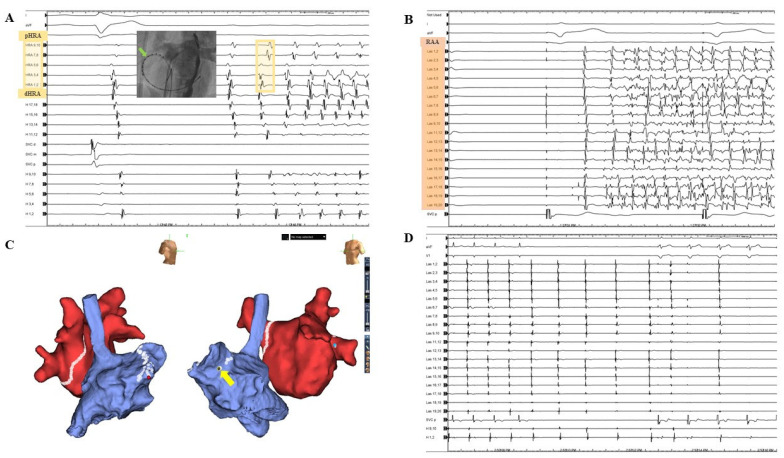
Additional ablation of the trigger site at RAA neck. Adenosine-induced atrial fibrillation (AIAF) originates from the neck of the right atrial appendage (RAA). A 59-year-old man was diagnosed with paroxysmal AF. Complete pulmonary vein isolation (PVI) was performed, and after confirming inducibility with isoproterenol, 12 mg of adenosine was administered. (**A**) The earliest activation of AIAF was the high right atrium (HRA) near the RAA. The green arrow indicates the earliest HRA 5,6 during AF after adenosine administration. (**B**) After the circular mapping catheter was placed in the RAA, adenosine was administered again to confirm the trigger site for AIAF. The potential of the circular mapping catheter occurred earlier than that of the HRA during AF. (**C**) The triggered site of earliest activation was eliminated until AIAF did not occur. The yellow arrow indicates the termination of the AF. (**D**) AF did not occur even after adenosine administration; therefore, the procedure was terminated.

**Figure 4 jcm-11-05679-f004:**
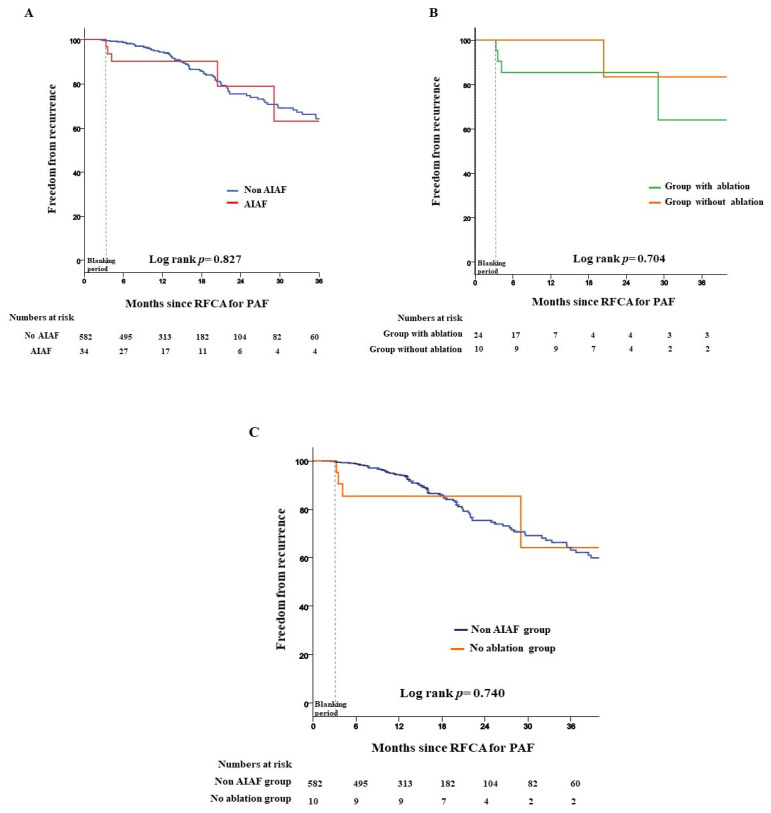
Kaplan–Meier analysis of freedom from recurrence. (**A**) Recurrence rate between adenosine-induced atrial fibrillation and no adenosine-induced atrial fibrillation. (**B**) In the AIAF group, the recurrence rate between the group with ablation for trigger site of AIAF and group without ablation. (**C**) Recurrence rate between non-adenosine-induced atrial fibrillation group and group without ablation. AIAF = adenosine induced atrial fibrillation; RFCA = radiofrequency catheter ablation; PAF = paroxysmal atrial fibrillation.

**Table 1 jcm-11-05679-t001:** Baseline characteristics of the study population. AIAF = adenosine-induced atrial fibrillation.

Variables	AIAF	Non-AIAF	*p*-Value
	(*n* = 34)	(*n* = 582)	
Age, years	53.9 ± 12.7	56.2 ± 10.9	0.305
Male, *n* (%)	28 (82.4)	440 (75.6)	0.370
Bodyweight (kg)	68.6 ± 10.9	71.6 ± 27.1	0.170
Height (cm)	169.6 ± 7.7	168.4 ± 8.8	0.420
Body mass index (kg/m^2^)	23.7 ± 2.6	25.2 ± 9.8	0.017
Congestive Heart Failure, *n* (%)	1 (2.9)	23 (4.0)	0.767
Hypertension, *n* (%)	9 (26.5)	208 (35.7)	0.271
Diabetes mellitus, *n* (%)	3 (8.8)	43 (7.4)	0.757
Previous stroke, *n* (%)	1 (2.9)	47 (8.1)	0.278
Vascular disease, *n* (%)	2 (5.9)	24 (4.1)	0.620
CHA_2_DS_2_-VASc score	0.9 ± 1.2	1.1 ± 1.1	0.195
Left ventricular ejection fraction (%)	57.4 ± 1.7	55.5 ± 4.8	<0.001
Left atrial diameter (mm)	37.0 ± 4.5	39.0 ± 5.3	0.014

**Table 2 jcm-11-05679-t002:** Cox-regression analysis for recurrence of atrial arrhythmia after RFCA.

Variable	Univariable		Multivariable	
	HR (95% CI)	*p*-Value	HR (95% Cl)	*p*-Value
Age ≥ 65 (years)	1.02 (0.65–1.60)	0.940		
Male sex	1.03 (0.66–1.63)	0.885		
Body mass index	0.99 (0.95–1.04)	0.778		
Congestive heart failure	2.63 (0.95–7.24)	0.062		
Hypertension	0.85 (0.57–1.28)	0.435		
Diabetes mellitus	0.93 (0.43–2.01)	0.853		
Stroke/TIA	1.20 (0.64–2.25)	0.562		
Vascular disease	1.33 (0.64–2.76)	0.438		
ADE induced AF	0.90 (0.37–2.22)	0.827		
LA diameter ≥ 40 (mm)	1.64 (1.12–2.41)	0.011	2.10 (1.36–3.27)	0.001
LVEF < 50 (%)	1.15 (0.64–2.06)	0.647		

Adjusted for age, sex, body mass index, congestive heart failure, hypertension, diabetes, history of stroke or TIA, vascular disease, ADE-induced AF, LA diameter, and LVEF. RFCA, radiofrequency catheter ablation; HR, hazard ratio; CI, confidence interval; TIA, transient ischemic attack; ADE, adenosine; AF, atrial fibrillation; LA, left atrium; LVEF, left ventricular ejection fraction.

## Data Availability

The data presented in this study are available on request from the corresponding author.

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
