# Peer review of "Clinical Significance of Adenosine-Induced Atrial Fibrillation after Complete Pulmonary Vein Isolation"

_jcm, 2022, doi:10.3390/jcm11195679_

Round 1
Reviewer 1 Report
Despite some limitations of the study and a recent similar publication, the topic is still of current interest with little available literature data.
Main Comments:
- A recent publication about the same topic has been recently published and the authors may cite it comparing their results in the Discussion (Ishimura et al. Long-term outcome of adenosine-induced atrial fibrillation after atrial fibrillation ablation: A propensity score matching analysis. Pacing Clin Electrophysiol 2022. doi: 10.1111/pace.14557).
- Please specify the exclusion criteria of study patients (redo-ablation?, valvular AF?) in the Methods.
- What was the rationale to perform linear LA ablation in paroxysmal AF patients?
- How many patients underwent a redo ablation during the follow-up?
- The retrospective, single center nature of the study should be mentioned in the Limitations.
- Any considerations about procedure and fluoroscopy times while performing AI-AF triggers mapping? Please discuss.
-Any direct correlation of the degree of adenosine induced bradycardia (pause cut-off) and the incidence of AI-AF?
Reviewer 2 Report
The authors present interesting study aiming to assess the prognostic value of atrial fibrillation induced by adenosine administration during radio frequency catheter ablation in regard to incident AF after pulmonary vein isolation. They did not find any association between adenosine-induced AF and recurrent AF regardless of wether additional ablation of a triggered site was performed or not.
My comments:
1) Introduction. To my opinion it is important to highlight the previous knowledge about the adenosine-induced AF. Were there previous studies assessing the percentage of occurrence, the link to dormant conduction, the association with clinical outcome?
2) Methods. If I get correctly the additional ablation was performed in all cases when dormant conduction was detected. It is not obvious from the text. I would suggest to emphasize this point.
3) Results:
- In the beginning the authors present baseline data in "the two groups". It is not clear what groups are meant. We сan guess that they are the groups with adenosine-induced AF and without it. But it should be highlighted in the text.
- The authors write that 70-80% of the patients showed negative dormant conduction. What is the exact percent?
- Additional ablation was performed in 10 of 34 patients. Why only in 10 of them? How were they selected for additional ablation (unspecified site in 17 of them, what is about the others?)
- Were those 8 patients with positive dormant conduction and AF among those 10 who underwent additional ablation?
- Did the author assess the prognostic value of positive dormant conduction in regard to recurrent AF?
- In multivariate Cox regression analysis, to my opinion, it is not appropriate to include both CHA2DS2-VASc score and all variables based on which this score is calculated.
- Figure 4 (B) and (C). Check the groups. It seems to me that they are mixed up. (At least under the curves numbers at risk are wrong). As a suggestion - combine these graphs and present the Kaplan-Meier analysis for three groups.
4) Discussion. To my opinion it is a little bit short. The main findings of the authors suggest that the pathophysiological mechanisms of adenosine induced AF and recurrent AF are different. Very important point is that patients with adenosine induced AF had smaller left atria (LA) than those without it and that LA > 40 mm independently predicted the recurrent AF. It means that in case of adenosine induced AF electrical properties of atrial myocardium play more important role in AF development than structural changes while in case of recurrent AF vise versa. It is worth to include in the discussion.
5) Limitation. In addition, due to paroxysmal nature asymptomatic AF might be missed and thereafter some cases of recurrent AF might be missed.
6) Conclusion. The authors write that AF most frequently occurred in RA, among these the most in the CT. In the results actually it is said that in 50% (the majority) the location in the RA was not specified.
